

# From dynamical localization to bunching in interacting Floquet systems

**Yuval Baum**[†], **Evert P. L. van Nieuwenburg**[†] **and Gil Refael**

Institute of Quantum Information and Matter, California Institute of Technology,
Pasadena, California 91125, USA

† These two authors contributed equally.

## Abstract

We show that a quantum many-body system may be controlled by means of Floquet engineering, i.e., their properties may be controlled and manipulated by employing periodic driving. We present a concrete driving scheme that allows control over the nature of mobile units and the amount of diffusion in generic many-body systems. We demonstrate these ideas for the Fermi-Hubbard model, where the drive renders doubly occupied sites (doublons) the mobile excitations in the system. In particular, we show that the amount of diffusion in the system and the level of fermion-pairing may be controlled and understood solely in terms of the doublon dynamics. We find that under certain circumstances the diffusion in 1D systems may be eliminated completely for extremely long times. We conclude our work by generalizing these ideas to generic many-body systems.


doi:10.21468/SciPostPhys.5.2.017

# 1   Introduction

Understanding properties of quantum many-body systems is a central theme in condensed matter physics. Already in one spatial dimension, many-body systems provide an enormous theoretical challenge. In recent years, the development of new numerical methods and the outstanding increase in computational power allowed us to peer into the many-body realm. Nevertheless, these methods are limited to low spatial dimensions and small system sizes.

A different direction for tackling the many-body problem lies within the framework of quantum control. The ability to manipulate and control many-body systems is a desirable goal. Influencing the interplay between different microscopic processes can dramatically reduce the level of complexity of these system and may shed light on their fundamental properties.

Amongst the promising means for achieving quantum control, periodic drives have drawn a great deal of attention over the last few years. Periodic drives emerged as a tool to control the band-structure and the dynamics of electronic systems *in situ*, both for solid-state setups and for cold atoms in optical lattices. These ideas have been demonstrated both for non-interacting and for interacting systems. In solid-state systems, Floquet engineering led to the emergence of exotic phases such as the non-interacting Floquet topological and Anderson insulators [1–8] and interacting time-crystals [9–15]. In cold atomic systems, periodic lattice-shaking techniques have been used to dynamically control tunneling [16, 17], induce a Superfluid-Mott transition [18] and generate artificial gauge fields [19–21].

Motivated by these ideas, we show in this work that quantum many-body systems may be controlled by employing a systematic driving scheme. We show that such a control gives rise to a plethora of phenomena ranging from a novel pairing mechanism and emergent composite particles to a complete elimination of diffusion. Before diving into the details, we summarize our main findings.

We propose a driving scheme under which many-body systems show an excitation-hierarchy. Particularly, we show that in the presence of the driving, the elementary particles in the system can be frozen while emergent composite particles become the stable mobile excitations. In models with interactions of range $M$, a hierarchy of different composite particles exists. Composite particles in a given hierarchy level, $R$, contain $R+1$ particles, where $R = 0$ corresponds to single particles.

We show that by systematically driving the system, one can eliminate (freeze) the composite particles at level $R$, rendering the particles at level $R+1$ the mobile units in the system. Such a driving scheme serves as a novel bunching mechanism for Fermions or Bosons in arbi-

trary dimensions. This mechanism can be easily understood for $M = 0$ (on-site interaction) or $M = 1$ (next-nearest-neighbor interaction), where the bunching mechanism is a real-space pairing mechanism, which renders doubly occupies sites (for $M = 0$) or neighboring sites (for $M = 1$) the mobile stable units in the system. In realizable cold atomic setups of Fermionic systems, such dynamical pairing mechanisms may lead to a buildup of superfluid correlations which cannot exist without the existence of the drive.

The above bunching mechanism sheds light on the fate of dynamical localization [22, 23] in the presence of interactions. Recently, it was shown in Ref. [24] that dynamical localization of spinless fermions does not survive the addition of nearest-neighbor interactions and the system becomes more and more diffusive as the interaction strength increases. The physical picture behind this becomes clear by considering the drive-induced bunching. While single particles remain localized, two neighboring particles behave as stable composite particles that become the mobile units in the system. We find the effective Hamiltonian for these composite particles and show that the original interacting Fermionic system behaves as a system of mobile hard-core Bosons. In particular, the revival of diffusion in the system may be understood (quantitatively) solely in terms of the composite particles' motion. Thus, we pinpoint the mechanism through which interactions destroy the localization.

One may wonder if the composite particles themselves may be localized as well. To answer that, we show that if a hierarchy level exits such that the composite particles are non-interacting, then these particles may be dynamically localized without generating higher order mobile composite particles. If such a scenario occurs, the many-body system cannot support particle diffusion. Indeed, such a scenario occurs for spinful fermions with on-site interactions, i.e., the Fermi-Hubbard model. Remarkably, such a non-diffusive state is not special to the standard 1D Fermi-Hubbard model, and it may be achieved also in the presence of additional hopping terms beyond the next-nearest-neighbor and in dilute systems in two or three spatial dimensions.

## 2 Background

### 2.1 Dynamical localization

We start by briefly reviewing the basic concepts of dynamical localization for non-interacting particles [22, 23]. In particular, we demonstrate how the dynamical properties of a system may be controlled by means of an external drive.

To that end, we consider a 1D lattice model in the presence of a time-dependent linear potential,

$$i\partial_t c_n(t) = H_{n-n'}c_{n'}(t) + E(t)n c_n(t) , \tag{1}$$

where $c_n$ annihilates a particle from lattice site $n$. The last term in Eq. (1) describes a uniform force, and therefore, may be described by a uniform time-dependent vector potential. In practice, the last statement is equivalent to the following unitary transformation,

$$\hat{U} = \exp\left(i \int^t dt' E(t') \sum_n n c_n^\dagger c_n\right), \tag{2}$$

and the transformed equation of motion is given by,

$$i\partial_t c_n(t) = H_{n-n'}e^{-iA(t)(n-n')}c_{n'}(t), \tag{3}$$

where $\dot{A}(t) = E(t)$ is the vector potential. The eigenstates of the discrete-translation-invariant Hamiltonian are labeled by their momentum $k$, i.e., $c_n^{(k)}(t) = e^{ikn - if(k,t)}c_k(0)$ where

$\dot{f}(k,t) = \mathcal{E}(k + A(t))$ with $\mathcal{E}(k)$ denoting the band-structure of $H$, and $c_k$ being the annihilation operator for particles with momentum $k$. As a result, the evolution of an initial state localized on a single site is then given by,

$$\Psi(t,n) = \sum_k \langle n | c_n^{\dagger(k)}(t) | \text{vac} \rangle = \int_{-\pi}^{\pi} \frac{dk}{2\pi} e^{ikn - if(k,t)}. \tag{4}$$

We say that the system is localized if the mean square displacement of generic localized initial states is finite at all times, i.e., $\sum_n |\Psi(t,n)|^2 n^2 < \infty$. We say that a system is exponentially localized if a finite $n_0 > 0$ exists, such that at all times $P(n,t) \equiv |\Psi(t,n)|^2 < e^{-\alpha|n|}$ for any $|n| > n_0$ and some $\alpha > 0$.

For Hamiltonians that include only nearest-neighbor hopping, with amplitude $J_0$, and a drive of the form $E(t) = E_0 \cos(\omega t)$, the probability for occupying site $n$ at time $t$ is given by,

$$P(t,n) = \left| \mathcal{J}_n \left( 2J_0 \sqrt{F_1(t)^2 + F_2(t)^2} \right) \right|^2. \tag{5}$$

where the functions $F_1(t)$, $F_2(t)$ are given by:

$$F_1(t) = \int_0^t \cos\left[ x \sin(\omega t') \right] dt' = \sum_n \frac{\mathcal{J}_n(x) \sin(n\omega t)}{n\omega}, \tag{6}$$

$$F_2(t) = \int_0^t \sin\left[ x \sin(\omega t') \right] dt' = \sum_n \frac{\mathcal{J}_n(x)(\cos(n\omega t) - 1)}{n\omega},$$

where $x = E_0/\omega$ and $\mathcal{J}_n$ are the Bessel functions of the first kind. In general, the argument of the sine and cosine functions, in the integral representation of Eq. (6), is $A(t')$ while the series-representation is specific for the cosine-drive.

In the limit $E_0 \to 0$ (no force) it is easy to see from the integral representation of Eq. (6) that $F_1 = t$ while $F_2 = 0$. Therefore, $P(n,t) = (\mathcal{J}_n(2J_0 t))^2$. Using the relation $\sum_n n^2 \mathcal{J}_n(z)^2 = z^2/2$, we get that $\langle n^2 \rangle = 2(J_0 t)^2$. The mean-square displacement increases to infinity and hence, without a force, the system is ballistic.

The limit $\omega \to 0$ corresponds to a constant force. Again, it is easy to see from the integral representation of Eq. (6) that,

$$P(t,n) = \left| \mathcal{J}_n \left( \frac{4J_0}{E_0} \sin\left( \frac{E_0 t}{2} \right) \right) \right|^2. \tag{7}$$

The mean-square displacement is bounded at all times and the system is exponentially localized. The above statement is nothing but the fact that lattice models with a linear potential give rise to a Wannier-Stark-ladder, in which all the eigenstates are localized. For a large enough force, such that $4J_0/E_0 \ll 1$, the initial state is practically frozen at its initial position. Equivalently, the probability in Eq. (7) is a manifestation of Bloch oscillations. The initial state returns to itself whenever $t = 2\pi/E_0 \times$ integer.

Finally, for both $E_0$, $\omega \neq 0$, it is instructive to separate the $n = 0$ term in the series representation of Eq. (6),

$$F_1(t) = \mathcal{J}_0(x)t + \sum_{n \neq 0} \frac{\mathcal{J}_n(x)}{n\omega} \sin(n\omega t) \equiv \mathcal{J}_0(x)t + v(t),$$

$$F_2(t) = \sum_{n \neq 0} \frac{\mathcal{J}_n(x)}{n\omega}(\cos(n\omega t) - 1) \equiv \mu(t).$$

Hence, the mean-square-displacement is given by $\langle n^2 \rangle = 2J_0^2 \left[ (\mathcal{J}_0(x)t + \nu(t))^2 + \mu(t)^2 \right]$. The functions $\mu$ and $\nu$ are bounded for all $t$ and $x$. As long as $\mathcal{J}_0(x) \neq 0$, the mean-square-displacement grows to infinity and the system is ballistic. Yet, for values of $x$ such that $\mathcal{J}_0(x) = 0$, the system becomes exponentially localized. At these values of $x$ the system effectively performs an integer number of Bloch oscillations every half period of the drive. Hence, dynamical localization is nothing but an extension to the notion of Bloch oscillations.

## 2.2 Floquet Hamiltonian of interacting particles

We wish to understand the fate of dynamical localization in the presence of interactions, where single particle band-structure cannot be defined. While the transformation in Eq. (2) eliminates the linear term also in the presence of interactions, the transformed Hamiltonian is interacting and cannot be solved by means of Fourier transform as before. Yet, for time-periodic drives, $E(t + T) = E(t)$, the Hamiltonian may be mapped into Floquet space, which sheds light on the the allowed processes and the relevant energy scales in the problem.

We first demonstrate the procedure for the non-interacting case. For time-periodic drives, Eq. (3) is invariant to time translations of $T = 2\pi/\omega$, and its solutions therefore have a Floquet form, i.e.,

$$c_n(t) = e^{-i\epsilon t} \sum_\gamma c_{n,\gamma} e^{i\gamma\omega t}. \tag{8}$$

Here, the operator $c_{n,\gamma}^\dagger$ creates a dressed state of $\gamma$ photons and a particle on site $n$. Inserting Eq. (8) into Eq. (3) yields a time-independent problem which is governed by the following Floquet Hamiltonian,

$$H^F = \sum_{n,n'} \sum_{\gamma,\gamma'} \mathcal{H}_{n-n',\gamma-\gamma'} c_{n,\gamma}^\dagger c_{n',\gamma'}, \tag{9}$$

where $\mathcal{H}_{x,\gamma}$ is the Fourier component of $H_x e^{-iA(t)x}$.

## 2.3 Driven Fermi-Hubbard model

Let us next see how a periodic drive can be used to control the Fermi-Hubbard model. Consider a one-dimensional Hubbard model of spinful fermions in the presence of a periodic drive which couples to the total fermion density. Such drives may be achieved by alternating electric fields in the case of charged fermions or by lattice shaking in the case of neutral fermions, see Fig. 1. Overall, the Hamiltonian is:

$$\mathcal{H} = \sum_{j,\sigma} (J_0 c_{j,\sigma}^\dagger c_{j+1,\sigma} + h.c.) + \frac{U}{2} n_{j,\uparrow} n_{j,\downarrow} + jF(t)n_{j,\sigma}, \tag{10}$$

where $c_{j,\sigma}^\dagger$ creates a fermion with spin $\sigma$ in site $j$, $n_{j,\sigma} = c_{j,\sigma}^\dagger c_{j,\sigma}$ is the density, $J_0$ is the hopping amplitude, $U$ is the energy cost of having a doubly occupied site, and $F(t)$ is a periodic function. In the following analysis we assume a cosine-drive, i.e, $F(t) = A\cos\omega t$. While the quantitative findings depend of the exact form of the drive, the qualitative result should not change.

Employing the transformation of Eq. (2), the Hamiltonian becomes,

$$\mathcal{H} = \sum_{j,\sigma} (J_0 e^{ix\sin(\omega t)} c_{j,\sigma}^\dagger c_{j+1,\sigma} + h.c.) + \frac{U}{2} n_{j,\uparrow} n_{j,\downarrow}. \tag{11}$$

where $x = A/\omega$. Employing the identity $e^{ix\sin(\omega t)} = \sum_m \mathcal{J}_m(x)e^{im\omega t}$, and transforming to the Floquet space yields the following Floquet Hamiltonian (summation over repeated indices):

$$H^F = J_0 \left[ \mathcal{J}_{m-l}(x) c_{j,\sigma,m}^\dagger c_{j+1,\sigma,l} + h.c. \right] + U n_{j,\uparrow,m} n_{j,\downarrow,m} + m\omega c_{j,\sigma,m}^\dagger c_{j,\sigma,m} \equiv H_J + H_U + H_\omega, \tag{12}$$

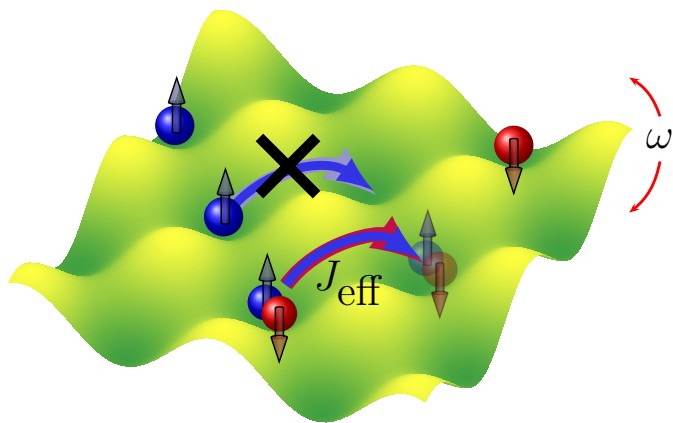

Figure 1: Spinful particles in a 2D shaken (driven) optical lattice. The particles' dynamics is governed by Eq. (10). In the presence of the drive, single particles become localized while doublons become stable and mobile units, with an effective hopping constant $J_{eff}$.

where the operator $c^{\dagger}_{j,\sigma,m}$ creates a dressed state of $m$ photons and a fermion with spin $\sigma$ on site $j$, $n_{j,\sigma,m} = \sum_l c^{\dagger}_{j,\sigma,l} c_{j,\sigma,m+l}$ and $\mathcal{J}_m$ is the $m$-th order Bessel function of the first kind.

While both formulations are equally hard to analyze, the form of Eq. (12) allows a better understanding of the allowed processes and relevant energy scales. The first term in Eq. (12), $H_J$, describes a process where a fermion hops to a nearest neighbor site while emitting or absorbing $m-l$ photons. The amplitude for these processes is given by $J_0 \mathcal{J}_{m-l}\left(\frac{A}{\omega}\right)$. The second and the third terms in Eq. (12), $H_U$ and $H_\omega$, can be thought of, respectively, as the energy cost for having doubly occupied sites (doublons) and photons in the system. We denote their sum by $H_0$. Notice that $H_0$ is diagonal in the configuration basis, i.e., $|\{n_{j,\sigma,m}\}\rangle$.

Eq. (12) will serve as the starting for analyzing which entities are mobile in the system.

# 3 Controlling the mobile units

## 3.1 Eliminating single particles

Let us begin by scrutinizing the motion of single particles. Are they dynamically localized also in the presence of interactions?

By setting the ratio $\frac{A}{\omega} = x_0$ to the first zero of the zeroth-order Bessel function, i.e., $\mathcal{J}_0(x_0) = 0$, we eliminate single particle processes that do not involve the emission or absorption of photons. The relevant energy scales in the problem are $\omega/J_0$ and $U/J_0$, and the only necessary requirement for single particle elimination is that $\omega$ is the largest energy scale in the problem.

In particular, for $\omega/J_0 \gg 1$, we may treat the hopping term, $H_J$, as a perturbation to $H_0$. The energy of a state under $H_0$ is given by the total number of photons and doublons in the system. Since $\omega$ is the largest scale in the problem, we expect the system to evolve within a photon number sector. We identify the photon number sector of the initial state as the zero-photons sector, and will now derive the effective Hamiltonian in that sector up to second order in perturbation theory. The derivation of the effective Hamiltonian follows the guidelines in Ref. [25]. A related derivation in the context of time-periodic Hamiltonians may be found in Ref. [26].

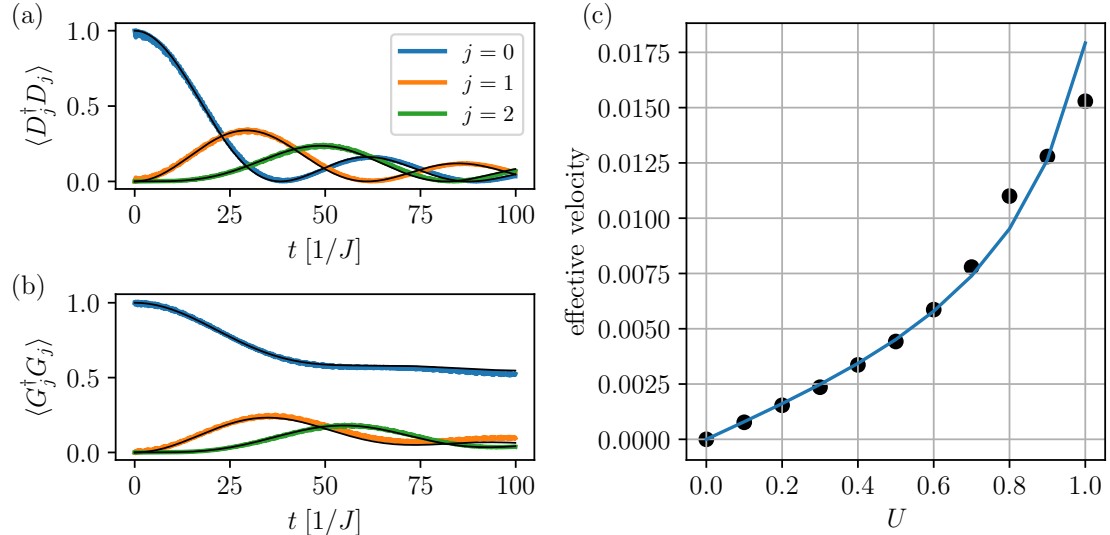

Figure 2: On the left, we show the density evolution of a single doublon (panel *a*) and of a single neighboron (panel *b*) initialized in the middle (site $j = 0$) of a 1D lattice. As time progresses, the probability for the doublon to remain at $j = 0$ or hop to the neighboring sites $j = \pm 1$ and $j = \pm 2$ follows the corresponding Bessel functions $|\mathcal{J}_j(2tJ_{\text{eff}})|^2$ (shown in black). This indicates that the doublons behave as single free units that do not break up into single particles, and validates the theoretical value of $J_{\text{eff}}$. A similar analysis can be performed for the neighborons, which shows that neighborons' mobility depends on their spin structure (see appendix 1). Panel (*c*) shows the analytical result for the effective velocity of neighborons (curve) in a spinless model with nearest-nieghbor interactions, in addition to the diffusion coefficient (divided by the mean-free-path) obtained in Ref. [24] (black dots).

Single particle states may be labeled by the occupied site and the number of photons, i.e., $|S_{j,N}\rangle$. With the notation $\langle S_{j',N}|H_J|S_{j,0}\rangle \equiv V_{j,j',N}$, the effective 1-particle Hamiltonian is given by:

$$\langle S_{j',0}|H_{\text{eff}}^{1p}|S_{j,0}\rangle = \langle S_{j',0}|H_0|S_{j,0}\rangle + V_{j,j',0} - \sum_{N,j''} V_{j',j'',N}^* V_{j,j'',N}\left(\frac{1}{E_i - E_N} + \frac{1}{E_f - E_N}\right)$$

$$\propto \sum_{N=-\infty}^{\infty} \frac{2(-1)^N |J_0 \mathcal{J}_N(x_0)|^2}{N\omega} = 0,$$

where $E_i$, $E_f$ and $E_N$ are the energies (under $H_0$) of the initial, final and intermediate states respectively. Similarly, the third order processes for single particles are identically zero, and hence we find that up to that order in $J_0/\omega$ the effective Hamiltonian for them is zero. In other words, single particle dynamics is completely frozen (dynamical localization, c.f. section 2.1.

## 3.2 Effective Hamiltonian for two particles

Next, we wish to find the effective Hamiltonian for two-particle states with no net spin (i.e. the particles have opposite spin). There are three types of such two-particle states: doublon states where the two particles share the same site, denoted $|D_{j,N}\rangle$; neighboron states where the particles reside in adjacent sites $j$ and $j + 1$, denoted $|G_{j,N}\rangle$, and singlon states, denoted

$|S_{j',j,N}\rangle$, where the two particles are separated by a distance $|j - j'| > 1$. In all notations, $N$ denotes the number of photons in the states with respect to the initial state.

The doublon-doublon matrix elements of the effective Hamiltonian are given by:

$$\langle D_{j',0}|H_{\text{eff}}^{2p}|D_{j,0}\rangle = U_{\text{eff}}\delta_{j,j'} + J_{\text{eff}}(\delta_{j,j'-1} + \delta_{j,j'+1}),$$

with $U_{\text{eff}} = (1 + 2\eta_+)U$ and $J_{\text{eff}} = \eta_- U$, where the dimensionless quantities $\eta_\pm$ are given by:

$$\eta_\pm = \left(\frac{J_0}{\omega}\right)^2 \sum_{N>0} \frac{(\pm 1)^N (2\mathcal{J}_N(x_0))^2}{(U/\omega)^2 - N^2}. \tag{13}$$

Neighboron-neighboron hopping is allowed with an effective hopping matrix element similar to the doublon case, and with zero on-site contribution. However, their spin structure affects their dynamics in an interesting way. Namely, their singlet component is completely localized while the triplet component is free to diffuse (see Appendix 1 for a more detailed discussion). For the remaining singlon-singlon case, we find that the matrix elements are all zero up to second order, i.e., $\langle S_{\tilde{j},\tilde{j}',0}|H_{\text{eff}}^{2p}|S_{j,j',0}\rangle = 0$. The same is true for the the neighboron-singlon and neighboron-doublon matrix elements.

The singlon-doublon matrix elements are not all zero up to second order. Assuming $K$ sites with periodic boundary conditions, each doublon state has non-vanishing matrix elements with six singlon states out of the total $K^2 - K$ non-doublon states. However, starting with a doublon state, the probability of finding any of the non-doublon states at a later time $t$, for finite $U$, is bounded by $\sim \eta_{-0}/U^2$, where $\eta_{-0}$ is the value of $\eta_-$ for $U = 0$ and $U$ is in units of $J_0$. In particular, the cumulative probability is given by,

$$\int_0^t dt' P(D \to S, t') = \int_0^t dt' \sum_{j',j''} \left|\langle S_{j',j'',N}|e^{iH_{eff}^{2p}t}|D_{j,0}\rangle\right|^2 \lesssim \frac{\eta_{-0}}{U^2}t \equiv \frac{t}{\tau}, \tag{14}$$

where we defined the stability time $\tau = \eta_{-0}/U^2$. In the relevant parameter regime, the stability time is long. For example, for $\omega/J_0 = 20$, $\eta_{-0} \sim 10^{-3}$ and thus, for $5 < U/J_0 < 20$ the stability time is $\tau \sim 10^5/J_0$. Hence, for times $t \ll \tau$, we can neglect the singlon-doublon matrix elements. Additionally, at long times, the contribution of these rare processes to the diffusion and transport properties is negligible compared to the fast process of doublon motion.

To summarize this part, we find that, up to second order, single particles are frozen while doublons and neighborons are two decoupled stable excitations that evolve non-trivially.

Beside the constraint forbidding two doublons from residing on the same site, doublons behave as free particles and the frozen single particles are transparent to the doublons (see Appendix 3 for a pathological case in which the doublon is trapped). Both the doublons and neighborons otherwise behave as free hard-core particles. For simplicity, we focus on the doublons dynamics from now on. As mentioned, neighborons behave in a slightly different way (c.f. Appendix 1).

The effective doublon Hamiltonian is thus

$$\mathcal{H}_{\text{eff}} = \sum_j (J_{\text{eff}} D_j^\dagger D_{j+1} + h.c.) + U_{\text{eff}} D_j^\dagger D_j, \tag{15}$$

where $D_j^\dagger$ creates a doublon on site $j$. The doublon operators fulfill the hard-core-boson relations, i.e., $[D_j^\dagger, D_{j'}^\dagger] = [D_j, D_{j'}] = 0$ and $[D_j, D_{j'}^\dagger] = (1 - 2N_j^D)\delta_{j,j'}$ with $N_j^D$ being the number of doublons in site $j$.

Employing a similar procedure to a square-wave-drive (rather than cosine) with amplitude $A$ and frequency $\omega$ leads to similar results with only quantitative differences. The condition for single-particle localization becomes $A/\omega = I$, where $I$ is an even integer, and the doublon Hamiltonian is identical to the cosine case up to the replacement of both $\eta_{\pm}$ by $\eta_{\text{SW}}$, which for a given $I$ is,

$$\eta_{\text{SW}} = \left(\frac{J_0}{\omega}\right)^2 \frac{2}{T\pi^2} \sum_{N \in \mathcal{P}_I} \left(\frac{N}{N^2 - I^2}\right)^2 \frac{1}{(U/\omega)^2 - N^2}, \tag{16}$$

where $\mathcal{P}_I$ is the set of all positive even integers for odd $I/2$ and positive odd integers for even $I/2$.

## 4  Discussion

### 4.1  Diffusion in the system

In the previous section we concluded that single particles are frozen while doublons are free to move, independently of the exact form of the drive.

In particular, since the doublons dynamics obey a simple tight-binding Hamiltonian, if a doublon is initially prepared in site $j = 0$, the probability of finding the doublon in site $j$ is $P(j,t) = |\mathcal{J}_j(2J_{\text{eff}}t)|^2$. For a square drive, this means that the system becomes ballistic with an effective velocity $2J_{\text{eff}} = 2\eta_{\text{SW}}Ut$. For $U/\omega \ll 1$, the function $\eta_{\text{SW}}$ is approximately $U$-independent, thus, the effective velocity is linear in $U$. As shown in Figures 2a and 2b, this theoretical prediction fits perfectly with the simulated doublon (with $\eta_-$ instead of $\eta_{\text{SW}}$) and neighboron dynamics.

Next, we compare our analytical results for the effective hopping (velocity) of neighborons with the time-dependent diffusion-coefficient obtained in Ref. [24], where it was calculated from the mean-square displacement (MSD) in a system of 15 spinless fermions on a 31 site chain. In this case we repeat our analysis for spinless model where the neighborons behave as mobile and stable particles with an effective hopping amplitude as above. Unlike in the spinful case, neighborons are interacting particles. While in dilute systems we expect the ballistic dynamics to last for long times, in dense systems a diffusive behavior is expected to arise at times much shorter than the stability time $\tau$. In a half-filled system of particles with nearest neighbor interactions, the average mean-free-path is expected to be of the order of a single lattice site, and hence we expect a good quantitative match between $D$ in Ref. [24] and the effective hopping calculated in our model. Indeed as is shown in Fig. 2c we find an excellent match between the analytical result $v = 2J_{\text{eff}}$ as computed for the neighborons and the diffusion coefficient extracted from the MSD in Ref. [24].

The analysis above is based on perturbation theory. However, the qualitative results holds to any order for times shorter than $\tau$. While the single-particle localization length depends on the parameters, the actual single-particle localization survives up to long times $t \gg \tau$. Similar statements can be made for the doublons. While the actual value of the effective parameters and the stability time $\tau$ depend on $J_0/\omega$ and $U/\omega$, the qualitative results remain unchanged as long as $U$ is not an integer multiple of $\omega$. Higher order processes lead to non-trivial evolution of singlon states, however, these processes affect the dynamics only for times much longer than $\tau$. It is safe therefore to neglect their effect on diffusion in the system. The effect of these higher order processes is addressed in Appendix 2.

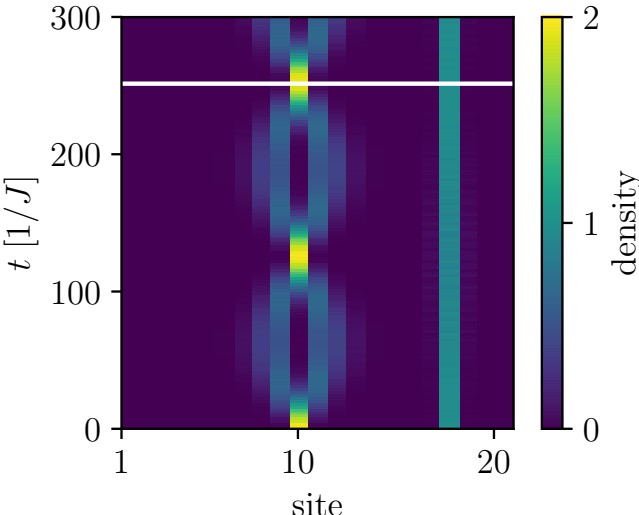

Figure 3: The density evolution of a single doublon initialized at site 10 and a single-particle initialized at site 18. In addition to the AC drive (lattice shaking) there is a small DC drive (lattice tilt), $E_0 = J_0/40$. While the single particles remain frozen the doublon performs Bloch oscillations with half the period of that of a single particle (indicated by the white line), due to its effective double charge/mass.

## 4.2 Doublon localization

We found that for any $U > 0$, dynamical localization is ruined and the system becomes delocalized due to the free motion of doublons. One may wonder whether the doublons themselves may be localized. Since the doublons in 1D are non-interacting, it is possible to both dynamically localize them by an alternating field, or 'Bloch localize' them by a uniform field. Both procedures destroy neighboron motion and high order singlon processes as well.

The second option can be achieved by adding to the original Hamiltonian, Eq. (10), a uniform force term, i.e., $\sum_j F_0 j n_{j,\sigma}$. While single particles experience a uniform force of $F_0$, doublons experience a uniform force of $2F_0$ and are expected to Bloch oscillate with double the frequency. Indeed, this is the case as shown in Fig. 3.

Dynamical localization may be achieved by driving the interaction term in Eq. (10), i.e., $U \to U_0 + A_2 j \cos(\omega_2 t)$. This translates to the following effective doublon Hamiltonian, $\mathcal{H}_{\text{eff}} = \sum_j (J_{\text{eff}} D_j^\dagger D_{j+1} + h.c.) + (U_{\text{eff}} + A_2 j \cos(\omega_2 t)) D_j^\dagger D_j$, which displays dynamical localization if $x_2 = A_2/\omega_2$ is tuned to a zero of $\mathcal{J}_0$. We have confirmed numerically that indeed this is the case.

We hence find that in both cases (uniform field and alternating field) the original interacting system does not support diffusion as long as the effective doublon-Hamiltonian is a valid description of the system, i.e., for $t \ll \tau$. At least for the uniform field case, the last statement is true to all orders in perturbation theory since the doublons become localized (due to the Stark-effect). Since the doublon stability is not exact, the absence of diffusion is not complete even in the presence of a uniform field. However, for times $t \ll \tau$, we expect the deviations from the non-diffusive states to be small.

## 4.3 Dynamical pairing

The elimination of single-particles and the emergent stability of the doublons acts as a controllable dynamical pairing mechanism. As we showed previously, for relatively long times,

the dynamics in the interacting fermionic system may be described solely by a hopping Hamiltonian of hard-core bosons. We show in the next section that the above statement holds also in higher dimensions. Unlike repulsive fermions (in the absence of phonons), hard-core particles in dimensions $d > 1$ are expected to show superfluid transition at low temperature. In particular, the critical temperature is expected to be of the order of $J_{eff}$. Below that temperature, one may expect to observe a buildup of superfluid correlation in the system. This type of physics may be realized in cold atomic setups where the buildup of superfluid correlations below $T = J_{\text{eff}}$ may be observed. The value of $J_{\text{eff}}$ may be controlled by the drive. In particular, it can be made of the order of 10% of the bare single particle hopping amplitude $J_0$. For shallow optical lattices, the hopping amplitude may exceed $50nK$. Thus, superfluid correlations may appear at $T \sim 5nK$, which is within the current experimental capabilities.

# 5  Extensions to general Hamiltoninas

In this section we highlight three different possible extensions: higher spatial dimensions, longer range hopping terms beyond nearest neighbors and a longer range of interactions.

## 5.1  Higher dimension

The above model can be trivially extended to higher dimensions. Starting with the d-dimensional version of Eq. (10),

$$\mathcal{H} = \sum_{\mathbf{j},\ell,\sigma} J_0(c^\dagger_{\mathbf{j},\sigma} c_{\mathbf{j}+\mathbf{r}_\ell,\ell,\sigma} + h.c.) + \frac{U}{2} n_{\mathbf{j},\uparrow} n_{\mathbf{j},\downarrow} + A\mathbf{j}\cos(\omega t) n_{\mathbf{j}\sigma}, \qquad (17)$$

where $\mathbf{j}$ is a d-dimensional vector denoting the lattice sites, $\mathbf{r}_\ell$ are the nearest neighbors vectors and the alternating force has an equal component along each lattice direction. Repeating the same procedure as in the 1D case, we find that for $A/\omega = x_0$ and up to second order in $J_0/\omega$, single particles are frozen while doublons behave as hard-core bosons with the following effective Hamiltonian,

$$\mathcal{H} = \sum_{\mathbf{j},\ell} J_{\text{eff}} D^\dagger_{\mathbf{j}} D_{\mathbf{j}+\mathbf{r}_\ell} + U_{\text{eff}} D^\dagger_{\mathbf{j}} D_{\mathbf{j}}, \qquad (18)$$

where $J_{\text{eff}}$ and $U_{\text{eff}}$ are identical to one-dimensional case. Similar to the 1D case, the addition of a uniform force or linear field leads to single-doublon localization. Yet, unlike the 1D case, hard-core bosons in $d > 1$ cannot be mapped to spinless fermions and cannot be considered non-interacting. While we expect a non-diffusive behavior in the dilute limit, at finite densities a diffuse behavior may arise due to the presence of interactions.

The most promising experimental candidates for observing the above effects are cold atom experiments. Hamiltonians such as Eq. (17), in $1-3D$, may be realized by means of optical lattices. The linear potential is then a tilt in the optical lattice and the oscillating force is equivalent to shaking the lattice. Shaking frequencies may exceed 20KHz, which for typical hopping amplitudes lead to $\omega/J_0 \sim 20$. In these systems we expect to see single-particle localization in all dimensions. In particular, if doublons are the only mobile degree of freedom in the problem, buildup of superfluid correlations at low temperatures are expected. A measurement of such correlations in a Fermionic system is clear evidence of both pairing and the ability to generate a highly controllable dynamical pairing mechanism.

## 5.2  Longer range hopping

In the above examples we consider a simple cosine band. However, more complicated band-structures may also be considered. For example, longer range hopping terms such as

$\sum_{j,m} \chi_m c_j^\dagger c_{j+m} + h.c.$ with $m \geq 1$ may be added to the original Hamiltonian. For a cosine drive, in order to dynamically localize single particles a multi-frequency drive is then needed. The hopping of range $m$ is localized by a drive of the form $A_g \cos(\omega_g t)$ where $m A_g/\omega_g = x_0$ with $x_0$ a zero of $\mathcal{J}_0$. In presence of different hopping terms, a combination of the above drives is needed. On the other hand, for a square-wave drive, it is possible to dynamically localize different hopping terms with a single frequency drive.

## 5.3 Longer range interaction

Consider the Fermi-Hubbard model with both on-site interactions and nearest-neighbor (NN) interactions,

$$\mathcal{H} = \sum_{j,\sigma} (J_0 c_{j,\sigma}^\dagger c_{j+1,\sigma} + h.c.) + A_0 j \cos(\omega_0 t) n_{j,\sigma} + (U_0 + A_1 j \cos(\omega_1 t)) n_{j,\uparrow} n_{j,\downarrow} + V n_j n_{j+1}, \tag{19}$$

where $n_j = n_{j,\uparrow} + n_{j,\downarrow}$. For $V = 0$, single particles and doublons may be localized by setting both $X_i = A_i/\omega_i$ to be a zero of $\mathcal{J}_0$, leaving no mobile excitation in the model. Turning on $V$ leads to the emergence of new mobile excitations in the problem. The new composite particles, neighborons, are composed of two particles that reside on neighboring sites. Unlike the neighborons in the previous section, these neighborons are completely diffusive. Higher numbers of particles that reside on neighboring sites can be understood in terms of the simple neighborons, and hence we do not consider them as new particles. The calculation of the effective Hamiltonian is identical to the calculation of the doublon effective Hamiltonian. The neighborons have an on-site energy of $V$ and a finite hopping amplitude that is proportional to $V$ (for small $V/\omega_0$). Unlike the doublons however, neighborons have NN interactions. Therefore, they can not be dynamically localized without generating other mobile composite particles. Indeed, driving the $V$ term, i.e., $V \to V_0 + A_2 j \cos(\omega_2 t)$ leads to neighborons localization if $A_2/\omega_2$ is tuned to a zero of $\mathcal{J}_0$. However, since neighborons are interacting, neighboring neighborons then become mobile.

The above procedure may be extended to a general interaction range. At each stage, it is possible to localize the relevant composite particle by driving the appropriate interaction term. The dynamical localization of interacting composite particles leads to higher order composite particles that become the mobile particles in the system. However, if at some stage the composite particles are non-interacting, c.f. the doublons in the 1D Fermi-Hubbard model, they may be localized *without* generating higher order mobile excitations. Notice that if a composite particle contains $R+1$ elementary particles, the leading order in the hopping amplitude of that composite excitation scales as $(J_0/\omega)^R$. Hence, even if the diffusion can not be eliminated completely, it may be made parametrically small.

# 6 Summary

In this work we employed Flouquet engineering to generic many-body systems. We presented a driving scheme that allows control over the nature of mobile excitations and control the amount of diffusion in the system. In particular, we showed that for the standard Fermi-Hubbard model in $d$ dimensions, the application of a single drive leads to single-particle localization, rendering doublons and neighborons the mobile excitations in the system. We find that the diffusion in the system may be understood, quantitatively, solely by the doublon dynamics. Moreover, the elimination of single-particles and the emergent stability of the doublons acts as a controllable dynamical pairing mechanism which may be realized in cold atomic setups.

We concluded by generalizing to models with $M$-range interactions, showing that a hierarchy of excitations exists. By systematic driving, the mobile units can be localized rendering higher order composite particles the mobile units in the system. In particular, if at some stage the composite particles are non-interacting, such as in the 1D Fermi-Hubbard model, the composite particles may be localized without generating other mobile units. These lead to a generic (non fine-tuned) many-body system that does not support diffusion.

It has not escaped our attention that a complete absence of diffusion in generic many-body systems is closely related to many-body-localization. It is beyond the scope of this work to determine whether the driving scheme we proposed retains integrability in generic $d$ dimensional clean many-body case.

## Acknowledgments

E.v.N. gratefully acknowledges financial support from the Swiss National Science Foundation through grant P2EZP2-172185. G.R. is grateful to the the NSF for funding through the grant DMR-1040435 as well as the Packard Foundation. We are grateful for support from the IQIM, an NSF physics frontier center funded in part by the Moore Foundation.

## A   Appendix

### A.1   Appendix 1: Neighborons effective Hamiltonian

Unlike doublons, the neighborons dynamics is less trivial. While neighborons hopping is not zero, half of their degrees of freedom are localized, and there is a trace of their initial condition also in the limit $t \to \infty$. To understand that phenomenon, we derive the neighborons effective Hamiltonian up to second order. We denote a neighboron state $|G_{j,\sigma,N}\rangle$ to be a spin $\sigma$ particle in site $j$ and an opposite spin, $\bar{\sigma}$, particle in site $j + 1$. Where $N$ is the number of photons. Up to second order, the matrix elements between neighborons and singlons or doublons are zero, hence, the neighborons space is a closed space. The non-zero neighborons-neighborons matrix elements are:

$$\langle G_{j,\sigma,0}|H_{eff}|G_{j,\sigma,0}\rangle = \langle G_{j,\sigma,0}|H_{eff}|G_{j,\bar{\sigma},0}\rangle = 2\xi_+. \tag{20}$$
$$\langle G_{j\pm1,\sigma,0}|H_{eff}|G_{j,\sigma,0}\rangle = \langle G_{j\pm1,\sigma,0}|H_{eff}|G_{j,\bar{\sigma},0}\rangle = \xi_-,$$

where

$$\xi_\pm = \sum_N \frac{(\pm1)^N \mathcal{J}_N^2(x_0)}{U + N\omega}. \tag{21}$$

The neighborons Hamiltonian may be transformed to momentum space to yield a momentum dependent $2 \times 2$ matrix (spin space):

$$H_{eff}(k) = 2(\xi_+ + \xi_- \cos k)(I + \sigma_x) \equiv f(k)(I + \sigma_x), \tag{22}$$

where $I$ and $\sigma_x$ are the identity and $x$ Pauli matrices respectively. Since $\xi_+ > \xi_-$ for every $U$, the function $f$ is always positive, thus, the eigenvalues of $H_{eff}$ are $\epsilon_- = 0$ and $\epsilon_+ = 2f$ with the corresponding eigenvectors are $V_- = (1 \; -1)/\sqrt{2}$ and $V_+ = (1 \; 1)/\sqrt{2}$. Next, we initialize a neighboron with a given spin configuration in site $j = 0$, e.g., spin up in site $j = 0$ and spin

down in site $j = 1$. The evolution of that initial state is given by:

$$\Psi(t,j) = \int \frac{dk}{2\pi} \frac{1}{\sqrt{2}} \left( V_- e^{-i\epsilon_- t} + V_+ e^{-i\epsilon_+ t} \right) e^{ikj} \tag{23}$$

$$= \frac{1}{2} \begin{pmatrix} 1 \\ -1 \end{pmatrix} \delta_{j,0} + \frac{1}{2} \begin{pmatrix} 1 \\ 1 \end{pmatrix} e^{-4i\xi_+ t} \mathcal{J}_j(4\xi_- t).$$

Hence, the probability to find a neighboron in site $j$ at time $t$ is,

$$P(t,j) = \frac{1}{2} \left( \delta_{j,0} + \left( \mathcal{J}_j(4\xi_- t) \right)^2 \right). \tag{24}$$

The particle density at site $j$ has contributions from two different neighborons states, i.e., $\rho(t,j) = P(t,j-1) + P(t,j)$. In particular, the density in the initially occupied sites, $j = 0$ and $j = 1$, is given by,

$$\rho(t,0) = \rho(t,1) = \frac{1}{2} \left( 1 + (\mathcal{J}_0(4\xi_- t))^2 + (\mathcal{J}_1(4\xi_- t))^2 \right). \tag{25}$$

Both become $1/2$ when $t \to \infty$. Hence, while half of the particle density diffuses, the other half remain in the initial position. Indeed, the numerical time evolution agrees perfectly with these results (see Fig. 2b in the main text).

In practice, the singlet part of the initial wave function remains localized while the triplet propagates freely. Indeed, initializing a neighboron in a singlet state leads to no diffusion. Such a phenomena may be used as a singlet filter. After preparing a generic zero-spin states, the triplet part diffuses away leaving a singlet state on the initial position.

## A.2 Appendix 2: Effect of Higher order processes

In the main text we considered all the processes up to second order. In particular, we showed that for times smaller than the doublons stability time $\tau$, the system is described by free doublons. Moreover, the time scale of doublons hopping is much shorter than $\tau$, e.g., for $U/J_0 = 10$ it is shorter by a factor of $\sim 3000$, and hence the diffusion in the system is completely dominated by the doublons dynamics.

Singlon hopping is still not allowed in third order. The most dominant third order process is a non-zero singlon-neighboron matrix element. However, these matrix elements are sparse and the matrix elements themselves are extremely small (much more than a single factor of $J_0/\omega$). For example, for $U/J_0 = 10$ and $\omega/J_0 = 20$, the matrix elements are of the order of $10^{-5} J_0$. This overall yields a typical time scale for these processes of the order of $\tau_1 \sim 10^{-8}/J_0$. In forth order, the most dominant processes are singlon hopping and doublon NNN hopping. These matrix elements are not sparse, however, their magnitude is of the order of $10^{-8} J_0$ which again lead to typical time scales of $\tau_{2,3} \sim 10^{-8}/J_0$.

Overall, there is a clear separation of scales. The doublon dynamics time scale is of the order $\sim 10^{-1}/J_0$, their stability time scale is $\sim 10^{-5}/J_0$ and the time scale of next relevant processes is of the order $\sim 10^{-8}/J_0$. The doublon dynamics is by far the most dominant process. Moreover, the addition of a uniform field on top of the drive eliminates the high order processes as well.

## A.3 Appendix 3: Doublon trap

The frozen single particles are in general completely transparent to the doublons. The process by which a doublon moves past a single particle however, results in the single particle being displaced by two sites. In second order, this process can be illustrated on three sites

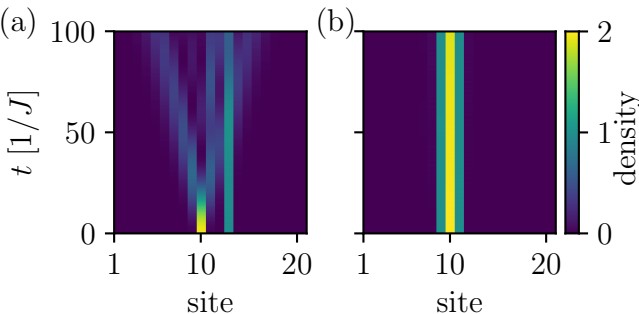

Figure 4: In panel *a*, a free doublon moves past a localized single-particle, thereby causing a displacement in the latter (it moves two sites left). In panel *b* we demonstrate that a doublon can be trapped due to this, when it is tightly surrounded by two single particles.

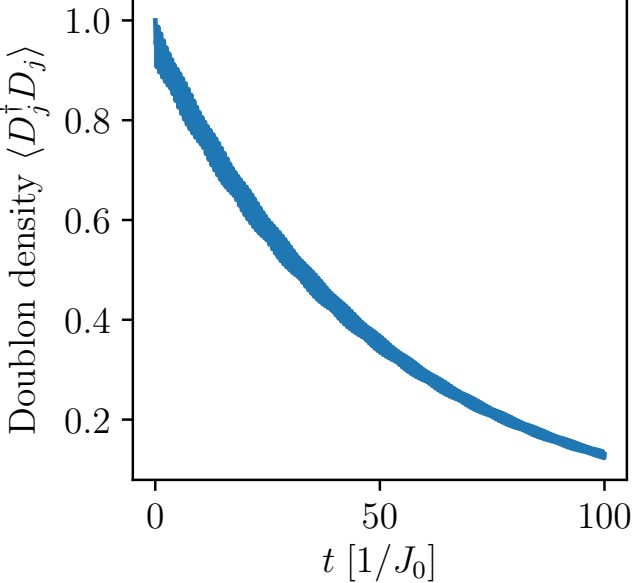

Figure 5: Decay of an initial doublon due to particle loss processes in the system, with $\gamma = 0.01$ in units of $J_0$.

as: $|\_\ \uparrow\downarrow\ \uparrow\rangle \rightarrow |\_\ \uparrow\ \uparrow\downarrow\rangle \rightarrow |\uparrow\ \_\ \uparrow\downarrow\rangle$. A numerical evaluation of this is shown in Fig. 4a. Hence, for the configuration in which a doublon is surrounded by two single particle states as $|\uparrow\ \uparrow\downarrow\ \uparrow\rangle$ (or different single particle spins for that matter), the doublon is completely trapped, c.f. Fig. 4b.

## A.4 Lindblad analysis

The dominant processes in cold atoms that cause the system to be imperfectly isolated from the environment are particle loss and dephasing. Such processes can be effectively described

using the Lindblad formalism for the time evolution of the density operator $\rho$,

$$\dot{\rho} = -i[H, \rho] + \frac{1}{2} \sum_\alpha \gamma_\alpha(t) \big( L_\alpha^\dagger \rho L_\alpha - \{L_\alpha L_\alpha^\dagger, \rho\} \big), \tag{26}$$

in which the jump operators $L_\alpha$ describe the system operator that is coupled to the environment, and $\gamma_\alpha(t)$ is the coupling rate.

We choose $\gamma_\alpha$ time independent, and thereby consider an effective infinite temperature environment that does not discriminate processes of different energy. In the presence of particle loss, doublons are clearly no longer stable. The main question then turns into a comparison of timescales, i.e. the effects we describe can be observed as long as the timescale for particle loss is much longer than the time of the experiment. For the parameters used in the main text, Fig. 5 shows the decay of an initial single doublon state in the presence of loss, i.e. $L_i = c_{i\sigma} + c_{i\bar{\sigma}}$ for all sites $i$, and $\gamma_i = \gamma = 0.01$.

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
