# Peer review of "From Dynamical Localization to Bunching in interacting Floquet Systems"

_SciPost Physics, doi:SciPost Phys. 5, 017 (2018)_

## Round 2 · Referee Report · Anonymous (Referee 1) · 2018-5-19

Strengths

1) Interesting topic & careful study 2) Discussion on interaction effects on dynamical localization. This is an important aspect of Floquet engineering in interacting systems 3) High quality of the discussion 4) Analytical arguments supported by few-particle simulations

Weaknesses

see report

Report

The paper investigates the physics of dynamical localization in the presence of interaction. The authors show how in a regime where single particles are localized due to a fine-tuned Floquet driving term, two- and more particle excitations can still propagate. Importantly, the authors identify a long time scale tau, below which the localized single particle dynamics separates from the delocalized two particle dynamics. The paper also investigates how two-particle excitations are localized and generalizes the 1d results to higher dimensions and generalized Hamiltonians.
Overall this is a careful, high-quality study of an interesting problem. It addresses genuine interaction effects in the field of “Floquet engineering”.
I have, however, two remarks/questions which may require some modifications of the paper

1) The authors use several times the phrase “non-interacting hard core bosons”. While in 1d hard-core bosons can indeed be mapped to non-interacting fermions, hard core bosons in higher dimension actually are strongly interacting (they have a non-trivial scattering cross section and are, e.g., diffusive for finite temperature and densities rather than ballistic). I would or example expect that the system described by Eq (18) shows a non-trivial two doublon dynamic even when single-doublons are localized dynamically. I would recommend to point out the difference of 1d and higher d in this respect and avoid the term “non-interacting hard core bosons” in higher dimensions completely.
2) The authors claim that for a static linear doublon potential there is no diffusion “to all orders in perturbation theory”. This statement is not explained and perhaps misleading (depending on the precise definition of “diffusion”). Even in the presence of a linear potential a generic interacting system will show some sort of subdiffusion, see e.g. arXiv:1101.4508

Minor remarks: Below Eq. (12) “The last term…” should probably be replaced by “The first term…”. Below Eq. (18): The authors write “As in the 1D case, the non-diffusive phase is not fine tuned”. I guess this statement refers only to the static and not the dynamics case.

Requested changes

see report

---

## Round 2 · Referee Report · David J. Luitz (Referee 2) · 2018-5-28

Strengths

Interesting and relevant topic.
See report.

Weaknesses

1) Not very clear notation 2) Numerics only for dilute case 3) Incomplete list of references 4) Confusing discussion of transport

Report

Disclaimer: I am one of the authors of the numerical study of transport in interacting Floquet systems Ref. [22, SciPost Phys. 3, 029 (2017)], which is challenged in the discussion of the present paper.

The authors present a very interesting study of the fate of dynamical localization in periodically driven systems in the presence of interactions. For such systems, it was shown recently in the numerical study in Ref. [22, SciPost Phys. 3, 029 (2017)] that interactions generically destroy localization and induce diffusive transport. In the present manuscript it is argued using perturbation theory that there are situations where dynamical localization can survive even in an interacting system. The argument is based on a perturbative construction of a hierarchy of increasingly complex mobile entities, identifying a criterion (constant offset of the oscillating electric field, or periodic drive of the interaction) under which not only the effective single particle states are localized due to dynamical localization, but also two particle entities, notably doublons or neighborons. The authors also present numerical simulations in support of their results as well as a comparison to the full many-body simulations of Ref. [22].

Overall, the results are very interesting and promising, since they also yield an analytical form for the effective hopping, which matches the diffusion constant calculated in Ref. [22], however there are several points in this study, which I think could be improved:

  • The list of references should be extended:
  • The references concerning discrete time crystals miss the pioneering work by Khemani, Lazarides, Moessner and Sondhi Phys. Rev. Lett. 116, 250401 (2016) and by Else, Bauer and Nayak Phys. Rev. Lett. 117, 090402 (2016).
  • The authors use a transformation of Floquet systems into a static problem, which to the best of my knowledge was first introduced by Jon H. Shirley Phys. Rev. 138, B979 (1965).
  • The presentation of the derivation of the main result is somewhat obscure and could be improved significantly. In particular it appears that some details and definitions are missing, thus making the derivation hard to follow. The main difficulty in the presentation of the section “II Background” seems to arise due to the repetition of the work of Jon Shirley (PR 138, B979 (1965)), leaving out important details. I suggest to refer to Shirley’s derivation and to follow his presentation to mitigate these problems. An example of the non standard form is e.g. Eq. (8), where it remains obscure where the photons come from and what the “Floquet form” is. In particular, Eq. (8) corresponds only to one solution of Eq. (3), whereas the Floquet theorem is a statement about the fundamental system of the differential equation, which one could for example get by writing the operator c_n (which is an LxL matrix) as a vector of length $L^2$.

  • The discussion of the results in Sec IV A should be clarified. As far as I understand it, within second order perturbation theory an effective model in Eq. (15) is derived, which is a simple tight binding model of doublons. Therefore, transport of doublons can only be ballistic. The authors write e.g. two lines after the heading B. Doublon Localization “and the system becomes diffusive due to the free motion of doublons”, which is clearly incompatible with Eq. (15).

  • My main critique is the discussion of the numerical results from Ref. [22]. These results show very clearly that the MSD, quantifying stroboscopic transport in a half filled, truly many-body driven system, grows linearly in time. It was shown in Ref. [22], that the domain of the linear growth grows also linearly with system size up to quite large system sizes, as predicted for clean diffusion, indicating that indeed bulk transport is probed and that the results do not suffer from transient boundary effects. Also note that the MSD is directly connected to the current-current correlation function and is therefore a direct measure of the nature of transport. The authors argue that transport in the same system (spinless fermions, which are discussed only in the appendix), is in fact ballistic and was incorrectly interpreted as diffusive due to finite size effects. This statement seems to be in conflict with the numerical evidence. Moreover, neither in the main text nor in the appendix the authors present the equation of motion for the neighborons, which is relevant for the system in Ref. [22]. If this equation contains interactions between the neighbourons, generically diffusion would be expected. In the numerical data of Ref. [22], a clear ballistic regime exists for very short times (constant as a function of system size), corresponding to a mean free path of one or two lattice sites, as expected in a half filled system, before the asymptotic diffusive transport kicks in. Using the assumption that the bulk transport in Ref. [22] is instead ballistic (contradicting numerical evidence), the authors equate the diffusion constant found numerically in Ref. [22] to the ballistic neighboron velocity, yielding the stunning correspondence presented in Fig. 2. At this stage, it remains unclear to me how this correspondence occurs, since the logic presented is not only in conflict with the numerical evidence, but also not self-consistent (e.g. in ballistic systems the mean free path has to be larger than the system size). This is apparent from the figure label in Fig. 2c, where an effective velocity (solid line, result of this paper) is compared to a diffusion constant (points, numerical result from Ref. [22]). Therefore it seems that some argument is missing in the logic to explain the good agreement with numerical simulations. Such an argument could be an analytical calculation of the mean-free path, which given the good agreement between the velocity and the diffusion coefficient, should be of the order of unity.

  • In the present work, two models are considered, most crucially the driven Fermi-Hubbard model (FHM) as well as the model of driven spinless fermions studied in Ref. [22]. The authors made an effort to separate the discussions of the two models by moving the spinless fermions model to the appendix., However they still compare their results to the results of Ref. [22], while the remaining results are for the behavior of doublons in the FHM. This leads to an intertwined discussion of the two models in Sec. IV, which is somewhat confusing. I suggest that the two models are strictly separated and not discussed in parallel to make it more clear. In particular, the effective neighboron Hamiltonian for spinless fermions should be stated explicitly, since this is the model which is compared to the numerical results.

  • The authors present numerical data confirming the localization of doublons according to their perturbative analysis. However, they limit their simulations to the dilute case of only 3-4 particles in the lattice. In this case, they find that the doublon lifetime is very long, justifying the analysis. For the strongly interacting limit, the case of half filling is more relevant and was considered in Ref. [22]. It is known from an analysis of the lifetime of doublons (e.g. [Phys. Rev. Lett. 104, 080401 (2010)]), that in the case of half filling and an interaction energy of the order of the kinetic energy this time is rather short. In this limit, it is not clear if the analysis in this paper is relevant for long enough times.

  • In summary, these arguments also cast doubt on the claim that the localization of doublons survives at higher filling and is a disorder free mechanism of localizing a many-body system.

In conclusion, I think that this paper contains very interesting and important results, which are surely relevant for the dilute case. Furthermore, the very good agreement of the analytical calculation with the numerical data in Ref. [22] suggests that a deeper insight can be gained also in the dense regime. However, the current form of the discussion contradicts exact numerical simulations for large many-body systems, suggesting that there is maybe a missing argument to explain this good agreement in Fig. 2c, which could maybe be fixed by focussing on the analysis of the diffusion constant and the mean free path in the dense case. The analytical part of the article could benefit from a more detailed and clear presentation.

I cannot recommend publication of the paper in the present form, due to the apparent inconsistency in the discussion of the results. However, I am confident that the authors will be able to improve the discussion along the above comments. I am convinced that this will lead to a deeper theoretical understanding of diffusion in driven systems and make it a valuable contribution to SciPost Physics.

Requested changes

1) Correct discussion of transport 2) Improve presentation of analytical result 3) Separate discussion of spinful and spinless models

---

## Round 3 · Referee Report · David J. Luitz · 2018-7-13

Report
The authors have addressed the comments raised by both referees in a satisfactory manner. The new version of the manuscript contains a discussion of transport for doublons and neighborons consistent with numerical findings.
I recommend publication of the revised manuscript in SciPost Physics.
Requested changes
There seem to be a few language glitches:
1) Abstract second to last line should read "in 1D systems".
2) 6 lines above the three line equation in Sec IIIA should read "A related derivation in the context..."

---

## Round 3 · Author Response

Dear Editor and Referees,
We are grateful for your efforts in reviewing our paper for submission. We especially thank the referees for their constructive comments and assesment of our manuscript. In the following, we believe we answer all comments made and concerns raised by the referees, and list all the changes we made in the revised version of our paper.
Sincerely yours, Yuval Baum, Evert van Nieuwenburg and Gil Refael
Response to Referee 1
The referee raised two concerns.
-
The referee wrote, ``The authors use several times the phrase “non-interacting hard core bosons”...hard core bosons in higher dimension actually are strongly interacting''.
The referee is certainly correct. Indeed, hard-core bosons in $d > 1$ cannot be mapped to spinless fermions and cannot be considered non-interacting. In all dimensions, the addition of a uniform force (linear field) leads to single-doublon localization. However, while we expect a non-diffusive behavior in the dilute limit, for $d>1$ and at finite densities a diffuse behavior may arise due to the presence of interactions.
We make this point clear in the main text. Moreover, we omitted the term 'non-interacting hard core bosons' whenever $d>1$ is considered.
-
The other concern the referee raised is regarding the doublon localization, ``The authors claim that for a static linear doublon potential there is no diffusion “to all orders in perturbation theory”. This statement is not explained and perhaps misleading.''.
Indeed the statement above can not hold for all times and is valid as long as doublons are stable. If the effective doublon-Hamiltonian was the true Hamiltonian of the system, then the presence of a linear field leads to localization (due to the Stark-effect) of the doublons. This localization is exact and true to all orders in perturbation theory. Of course, the effective Hamiltonian is not exact and the stability time of the doublons is finite. Therefore, the absence of diffusion is not complete even in the presence of a uniform field. Yet, for times $t$ much smaller than $\tau$, where the effective doublon-Hamiltonian is a valid description of the system, we expect the deviations from the non-diffusive states to be small.
This is now also elaborated upon in the main text.
Response to Referee 2
First and foremost, we would like to emphasize that the comparison to Ref. (22) (now Ref (24)) was made not in order to challenge it, but rather to show the agreement between our analysis and their work. We think of our work as complementing the results in Ref. (22).
We wish to start with the main critique the Referee raised regarding the comparison with Ref. (22) (in the old version). After private communication with the authors of Ref. (22), we are certain that the transient diffusive behavior they observed is a true effect and it is not due to finite size effects. This was the only question we raised on Ref. (22), and was not meant as a challenge.
We revised the main text accordingly and wrote: ``\textbf{Unlike in the spinful case, neighborons are interacting particles. While in dilute systems we expect the ballistic dynamics to last for long times, in dense systems a diffusive behavior is expected to arise at times much shorter than the stability time $\tau$. In a half-filled system of particles with nearest neighbor interactions, the average mean-free-path is expected to be of the order of a single lattice site, and hence we expect a good quantitative match between $D$ in Ref. (22) and the effective hopping calculated in our model}''.
We would like to stress that the main point in our paper is that, surprisingly, dynamical localization breaks already at the extremely dilute case of a single doublon (spinful) or neighborons (spinless). Indeed, as the referee wrote, we support these statements by numerics of dilute systems. We rely on the impressive numerical effort of Ref. (22) to claim that our model can hold even beyond the dilute limit by showing that the diffusion constant calculated in Ref. (22) perfectly matches the hopping amplitude of the neighborons. While the mean-free-path is expected to be of the order of a single lattice site, it seems that it is identically equal to one. Since this is not the focus of our paper, we choose not to discuss the exact match which requires an exact calculation of the mean-free-path for a specific model.
The nature of our work is not numerical and we do not see the fact that we simulate only dilute systems as a weakness as the referee suggested. We believe that Ref. (22) and our work compliment and support each other. While the stability time in the spinless non-dilute case may be short, the comparison to Ref. (22) indicates that the validity of our simplified model may extend beyond the dilute limit.
Finally, by suggestion of the referee, whenever we refer to the spinless case we emphasize the differences with respect to the spinful case.
-
The referee is concerned with our Floquet derivation. In particular, he wrote: ``The authors use a transformation of Floquet systems into a static problem, which to the best of my knowledge was first introduced by Jon H. Shirley''.
The idea that a time-periodic function or Hamiltonian may be mapped into a static (infinite matrix) Hamiltonian can be traced back to seminal works of Floquet, Bloch and others (including Shirley). The growing field of Floquet physics has produced hundreds of papers, and common practice in the field is not to cite all these seminal works, most of which are considered textbook materials nowadays. Upon closer investigation however, it seems common in the cold-atom and ion-trap community to include this particular reference, and we have done so now too. We thank the referee for this suggestion.
-
The referee also wrote: ``...the non standard form is e.g. Eq. (8), where it remains obscure where the photons come from and what the 'Floquet form' is''.
Here we would beg to differ. Equation (8) in our paper is simply the Floquet theorem, stating that a solution to the time-periodic Schrodinger equation is a phase factor times a periodic function. The latter is written as a Fourier series. In fact, this particular form can be found in Eq. (3) of the above-mentioned reference to Shirley. We refer to this solution form as the 'Floquet form' and to the integers in the Fourier expansion as the 'photon number', according to common terminology. Hence Eqs. (8) and (9) were meant as a short recap of a heavily used formalism.
-
Finally, the referee wrote ``Eq. (8) corresponds only to one solution of Eq. (3)''.
This might be the origin of some confusion. All the solutions to Eq. (3) have the form of Eq (8). While, indeed, each eigenstate of the Floquet Hamiltonian corresponds to one solution of Eq. (3), all eigenstates are accessible by this procedure.
-
The referee wrote ``The authors write e.g. two lines after the heading B. Doublon Localization “and the system becomes diffusive due to the free motion of doublons”, which is clearly incompatible with Eq. (15).''
We thank the referee for pointing this out. The word diffusive is wrong in this context, and we've replaced it with the word 'delocalized'. In practice, for $t$ much smaller than $\tau$ we expect to observe a ballistic behavior (see the answer to Referee 1, and the list of changes).

---

## Round 3 · List of Changes

We omitted the term 'non-interacting hard core bosons' whenever $d>1$ is considered.
We added relevant references based on referee reports.
Page 6, after Eq. 18: we comment about the first point the referee made (see above).
Page 7, second column, second paragraph: we comment about the second point referee 1 made (see above).
Page 6, first column, second paragraph: we comment about the main point of referee 2 regarding the interacting nature of the composite particles in the spinless case and the expectation to find a diffusive behavior.
Below Eq. (12) ``The last term...'' was replaced by ``The first term...''.

---

## Editorial Decision

published